# Pharmacological Prevention of Hypersensitivity Reactions Caused by Iodinated Contrast Media: A Systematic Review and Meta-Analysis

**DOI:** 10.3390/diagnostics12071673

**Published:** 2022-07-09

**Authors:** Chen Hsieh, Shan Chia Wu, Russell Oliver Kosik, Yu-Chen Huang, Wing P. Chan

**Affiliations:** 1Department of Family Medicine, Taipei City Hospital, Heping Fuyou Branch, Taipei 100058, Taiwan; hsiehchen91@gmail.com; 2Department of Radiology, Wan Fang Hospital, Taipei Medical University, Taipei 116079, Taiwan; 107148@w.tmu.edu.tw; 3Department of Radiology, School of Medicine, College of Medicine, Taipei Medical University, Taipei 11031, Taiwan; russkosik1@hotmail.com; 4Research Center of Big Data and Meta-Analysis, Wan Fang Hospital, Taipei Medical University, Taipei 116079, Taiwan; 5Department of Dermatology, Wan Fang Hospital, Taipei Medical University, Taipei 116079, Taiwan; 6Department of Dermatology, School of Medicine, College of Medicine, Taipei Medical University, Taipei 11031, Taiwan

**Keywords:** contrast media, hypersensitivity, premedication, steroids

## Abstract

**Objectives:** Hypersensitivity reactions (HSRs) are uncommon but serious adverse events following the administration of iodinated contrast media (ICM) prior to CT imaging. While premedication is almost universally given in high-risk patients, there is a lack of evidence regarding the efficacy of such premedication. This study aims to determine the efficacy of premedication with corticosteroids prior to ICM administration in the prevention of HSRs through meta-analysis. **Materials and Methods**: An extensive review of the literature yielded 404 potentially relevant studies. Of these, five studies met the inclusion criteria of this meta-analysis. Pooled HSR event rates were obtained from each of the studies for both patients who had and who had not received premedication with corticosteroids. Heterogeneity between studies was also determined. **Results:** A total of 736 patients across all five studies were included in the analysis. Patients who did not receive premedication had initial pooled HSR rates of 0.16 (95% CI, 0.07–0.35) across all studies. Following premedication, pooled HSR rates dropped to 0.02 (95% CI, 0.01–0.06). Patients who had prior HSRs were significantly less likely to experience HSRs (OR = 0.09; 95% CI, 0.03–0.25; *p* < 0.00001) after treatment with premedication. **Conclusions**: This meta-analysis offers evidence for the reduction in the recurrence of moderate and severe HSRs in patients who have a history of such reactions. Premedication with corticosteroids prior to ICM administration is thus highly recommended in high-risk patients.

## 1. Introduction

The rapid increase in the use of contrast-enhanced diagnostic CT scans using intravenous iodinated contrast media (ICM) has coincided with a growing number of ICM-related hypersensitivity reactions (HSRs) [1]. HSRs are classified into immediate reactions (IHR), which occur within 1 to 6 h after ICM administration, and nonimmediate reactions (NIHR), which occur over 6 h after ICM exposure [2,3,4]. IHRs most frequently result in urticaria, sometimes accompanied by vomiting, diarrhea, and abdominal pain, but more rarely, they can also cause severe reactions such as anaphylaxis [2,3,4,5,6]. NIHRs commonly present with maculopapular exanthem, but in rare cases, they can result in severe skin reactions such as Stevens–Johnson syndrome, toxic epidermal necrolysis, and fixed drug eruptions [2,3,4,7]. The pathophysiology behind IHRs and NIHRs significantly differs: the former is in part associated with histamine and tryptase release from basophils and mast cells [8], although in most patients, the mechanism is nonallergic; the latter is T-cell-mediated hypersensitivity reactions [9,10,11,12]. While the precise mechanisms behind IHSRs are not fully understood, the osmolality, size, and complexity of the ICM appear to alter the risk of HSRs. Several reports have shown that nonionic ICMs confer less risk of HSRs than ionic ICMs [13,14,15].

The appropriate handling of contrast-related HSRs is an important issue in clinical practice. Premedication is believed to reduce the likelihood of immediate adverse events and is usually administered to patients who have experienced an HSR in the past [13]. Typically, premedication protocols include the administration of corticosteroids and/or antihistamines under a defined protocol that usually varies from institution to institution [16]. This protocol may also change depending on certain patient-specific factors, such as the urgency of the need for imaging or an inability to receive steroids for medical reasons [12]. Further, the specific drug administered may be affected by the patient’s condition. For example, a patient who is unable to take oral medications will typically receive premedication intravenously. Such variation has made it difficult to determine the exact efficacy of premedication, as well as identify the optimal premedication protocol. 

However, since the 1980s, premedication with corticosteroids has been a mainstay and has been shown to offer protection prior to the administration of certain ICMs [13,16,17,18,19,20]. However, the benefits of steroid premedication after low-osmolar nonionic ICM are debatable. Only a single randomized trial has demonstrated that premedication provided to average-risk patients prior to low-osmolarity ICM administration decreases the incidence of mild and aggregate immediate adverse events [21]. There is a lack of strong evidence confirming the efficacy of premedication in patients with a history of moderate or severe reactions. Although recommendations, along with specific regimens of corticosteroid prophylaxis, are included in The American College of Radiology (ACR) Manual on Contrast Media, actual clinical practices continue to vary considerably across institutions and even from physician to physician. In addition, multidisciplinary strategies require integration among different specialists, especially allergists and radiologists, in the prevention of adverse reactions [22].

The present study aims to systematically review the literature concerning corticosteroid premedication prior to ICM administration and to use meta-analysis to assess the effectiveness of prophylaxis in patients with a history of moderate or severe reactions.

## 2. Materials and Methods

We conducted this meta-analysis according to the Preferred Reporting Items for Systematic Reviews and Meta-Analyses (PRISMA 2020) [23].

### 2.1. Literature Search

We performed an electronic search on PUBMED, EMBASE, and the Cochrane database (1959 to July 2021) for human studies in English. Search terms included medical subject headings (MESH) and the keywords “premedication,” “contrast media,” and “hypersensitivity.” We also searched the references within the searched articles and related reviews. During the submission process, a further search was extended to May 2022, and the results are discussed in the limitations section.

### 2.2. Eligibility Criteria and Study Selection

The inclusion criteria were as follows: (1) Study subjects had prior ICM-induced HSRs; (2) the contrast media were low- or iso-osmolarity; (3) HSRs were classified as mild, moderate, or severe in accordance with previously published guidelines, and patients had at least moderate or severe HSRs; (4) steroid premedication was administered prior to subsequent ICM use; and (5) the study contained sufficient data to allow calculations of the parameters of interest. The exclusion criteria were: case reports, abstracts, duplicated reports, or studies based on a previously published study. Articles were screened by one author (C.H) according to the inclusion and exclusion criteria. Two other investigators (S.C.W and Y.C.W) independently evaluated the relevant articles. Discrepancies between investigators were resolved through consensus.

### 2.3. Data Extraction and Quality Assessment

For each included study, we extracted the following information: publication year, study period, country, study design, sample size, age, gender, contrast media type, dosage of premedication, and the number of HSRs. The selected studies were included as a case series based on our data extraction. A special tool for evaluating the methodological quality of case reports and case series was also used in the included studies [24].

### 2.4. Statistical Analysis

Extracted data were classified into initial HSR rate and premedicated HSR rate in the same group of patients. The pooled HSR event rates were the measures of interest. Both random- and fixed-effects models were generated, using the inverse variance weighting and DerSimonian–Laird methods, respectively. 

To evaluate the efficacy of premedication, we compared the HSR event rates in patients who were premedicated with steroids to the initial HSRs rates without premedication in patients with prior HSRs. None of the before-and-after studies that measured dichotomous outcomes reported data in a matched format; therefore, these outcomes were analyzed as unmatched data, which are shown to be similar and easier to interpret than matched analyses [25]. The effect sizes were calculated as odds ratios (OR) with corresponding 95% confidence intervals (95% CI). The Mantel–Haenszel random-effects model was adopted. Heterogeneity across studies was assessed using the I^2^ index. An I^2^ >50% was considered a significant heterogeneity. When significant heterogeneity was observed between studies, the random-effects model was selected as more appropriate. Publication bias was evaluated using the Egger’s regression test. Statistical analysis was performed using the R package metafor.

## 3. Results

### 3.1. Literature Search

A total of 404 studies were preliminarily identified. These initial studies were screened by titles and abstracts. Following screening, thirty-four studies remained for full text evaluation. Twenty-one studies were found not to fit the inclusion criteria, and seven studies were excluded due to insufficient data. One study published in 2022 qualified for the analysis, but it was not discussed as it was identified after the submission date of this article (Figure 1)

### 3.2. Study Characteristics and Quality Assessment

The five remaining studies included two Japanese studies, two Korean studies, and an American study [26,27,28,29,30]. The Japanese studies included 117 and 271 patients, respectively; the Korean studies included 30 and 116 patients, respectively; and the American study included 202 patients. This yielded a total of 736 patients, which included participants of all ages and both males and females. The studies were published between 2011 and 2017, and study periods ranged from 1 year and 6 months (the earlier Japanese study) to 9 years and 6 months (the American study). Following premedication, the Japanese studies reported one and two moderate and/or severe HSRs, respectively; the Korean studies reported one and nine moderate and/or severe HSRs, respectively; and the American study reported three moderate and/or severe HSRs. Additional study characteristics are shown in Table 1.

Because the data were extracted without a control group, all studies were included as a case series. Quality assessment was confirmed using a tool aimed at ensuring the methodological quality of case reports and case series. Eight items were evaluated, and these were categorized into four domains: selection, ascertainment, causality, and reporting. The eight items are shown in Table 2. Overall, the study quality was found to be satisfactory. 

### 3.3. Pooled Event Rate Estimates

A total of 736 examinations were performed on patients who received premedication before the use of contrast media. The initial moderate and severe HSR event rates ranged from 0.01 to 0.58, which accounted for the high inter-study heterogeneity (I^2^ index 95%). The random-effects pooled event rate for nonmedicated patients was 0.16 (95% CI, 0.07–0.35) across all studies (Figure 2). After premedication, the pooled event rate decreased to 0.02 (95% CI, 0.01–0.06) (Figure 3).

### 3.4. Effectiveness of Premedication

The total number of events in the premedication group was 16 compared to 155 in the group that did not receive premedication. A pooled data analysis revealed a significant difference between the two groups. (OR = 0.09; 95% CI, 0.03–0.25; *p* < 0.00001) (Figure 4).

### 3.5. Publication Bias

Publication bias was not assessed as there was an inadequate number of included studies to properly perform a funnel plot or the Egger’s regression test.

## 4. Discussion

The objective of this systematic review is to determine the efficacy of premedication in patients with prior moderate or severe HSRs. Our meta-analysis shows that patients who have had prior HSRs are significantly less likely to experience HSRs after treatment with premedication.

The overall incidence of acute HSRs using low-osmolality ICM is very low (0.2~0.7%) worldwide [1,31,32]. Serious HSRs are considered extremely rare in daily practice. In Taiwan, I-Hao et al. undertook a retrospective study revealing that the rate of HSRs was 0.66% for nonionic contrast media, with only two severe reactions amongst 24,979 patients [33]. In a large Japanese study, Katayama et al. reported a rate of severe HSRs of 0.04% for nonionic contrast media examinations [14]. In a multicenter Korean study, Cha et al. described a rate of moderate HSRs of 0.11% and a rate of severe HSRs of 0.01% amongst 196,081 patients [34]. Fatal reactions following ICM administration are even rarer. Katayama et al. found a rate of one fatality per 170,000 contrast media administrations, while Caro et al. identified a rate of 0.9 per 100,000 injections of low-osmolality ICM [14,35]. While routine premedication is not recommended due to the low incidence of HSRs, many experts believe that premedication in high-risk patients reduces the rate of HSRs, regardless of the high number needed to be treated [36,37].

To prevent HSRs in high-risk patients, several premedication strategies have been proposed, mostly involving a combination of steroids and/or antihistamines. Many consider steroids to have a fundamental role in both the prevention and treatment of moderate to severe HSRs, even though the effectiveness of steroids is controversial. Some studies have demonstrated a protective effect of steroids [30,36], while other studies have reported breakthrough reactions following the use of low-osmolar contrast agents, despite premedication with steroids [38,39]. More convincing research is needed to reach a consensus.

In this analysis, the type of steroids used included hydrocortisone, prednisolone, prednisone, and methylprednisolone. The total medication dosage ranged from 25 to 125 mg in prednisolone equivalents. The frequency of administration was based on the duration of the different steroids. Intravenous steroids were given to those patients who were unable to take oral medications as well as in urgent cases. Fundamentally, the regimens used adhered to the recommendations advised by the ACR Manual on Contrast Media. However, the small number of studies included in our analysis limits the ability of this study to identify potential changes in efficacy that may occur with varied dosing and administration strategies.

Changing the type of contrast media used may also play a role in the development of HSRs. Park et al., in a large study of 1178 patients, demonstrated that simply changing the contrast media, even without premedication, can result in a dramatic drop in the rate of recurrent HSRs, although changing the contrast media in addition to premedication is even more efficacious [39]. Both the European Society of Urogenital Radiology Guidelines [40] and the ACR Manual on Contrast Media advise using different classes of contrast media when patients have had a prior HSR. Nevertheless, the advantages of changing the type of contrast media, typically from one low-osmolar agent to another, are uncertain. In fact, Davenport et al. suggested that changing the class of contrast media yielded no significant difference when premedication was used [41]. Abe et al. demonstrated that replacing one low-osmolar, nonionic contrast agent with another can be effective when premedication is not used, though changing contrast agents reduces the effectiveness of premedication [30]. In our review, low- or iso-osmolarity contrast media were used in all studies. According to Abe et al., the rates of moderate and severe HSRs only improved slightly with premedication when the same contrast media were used [30]. However, Kim et al. reported that premedication was effective, showing that only 1 premedicated patient out of a group of 11 patients with prior severe HSRs experienced a severe HSR when the same contrast agent was used again [29]. Other studies incompletely documented the precise types of contrast media that were used. Therefore, the need for premedication when the class of contrast media is changed remains uncertain.

While the types of contrast agents and types of premedication may be important because the precise mechanisms of HSRs remain elusive, the optimal prevention strategies also continue to be elusive. New evidence suggests that HSRs may occur for a number of reasons and should be distinguished in at least two ways. While nonallergic HSRs are responsive to premedication, rarer allergic HSRs are not nearly as responsive [42]. These different types of HSRs may appear clinically similar, though probably occur through distinct inflammatory or allergic mechanisms and, therefore, likely require distinct treatment or prevention strategies. Furthermore, it may be possible to pre-emptively identify patients susceptible to allergic HSRs through skin or other testing. Such novel approaches towards HSRs may explain why dated methods of thinking about HSRs have not been as successful as clinicians would have liked. Moving forward, the identification of the precise mechanisms that lead to an HSR will be fundamental in developing adequate treatment and prevention strategies. 

Our study had several limitations. Firstly, this review did not include one article [43] in the results section because this analysis was completed and submitted before the article was published. It was a single-center, retrospective study conducted in America. McDonald et al. demonstrated the superiority of ICM substitution over steroid premedication with the same ICM in preventing all the severity of HSRs. The referential data provided 589 examinations using steroid premedication, and the moderate and severe HSR event rate decreased from 29.4% (moderate reaction: 150 of 589 examinations [25.5%]; severe reaction: 80 of 589 examinations [13.6%]) to 2.3% (moderate reaction: 13 of 589 examinations [2.2%]; severe reaction: 0 of 589 examinations [0%]). The result was similar to our analysis. 

Secondly, the strength of evidence in the articles included in this analysis was relatively low as all studies lacked control groups. While the current consensus, as stated in the ACR Manual on Contrast Media, calls for patients with a history of a severe hypersensitivity reaction to receive premedication, evidence from cohort studies or randomized, controlled trials to support this recommendation is lacking. A high inter-study heterogeneity of reported rates of initial and premedicated HSRs was observed, and it may be partially secondary to the varied races/countries, study designs, and the included number of patients.

Another important factor is the potential confounding effect of antihistamine administration. Typically, though not always, antihistamines are administered as a part of the premedication process. Because antihistamines and corticosteroids have distinct anti-inflammatory mechanisms, it is difficult to compare the efficacy of corticosteroids alone and/or the combination of corticosteroids and antihistamines. While both classes of drugs may play a role in the prevention of HSRs, the relative role of each is difficult to determine as is the optimal combination. In fact, Park et al. demonstrated that an antihistamine used as a premedication alone can significantly reduce the risk of HSRs [39]. Furthermore, because different regimens of premedication with both corticosteroids and/or antihistamines were used in each study analyzed, a comparison across the studies is limited. An ideal meta-analysis would use studies that all employed the same premedication regimen; however, the number of studies where the same regimen was used is insufficient at this time.

## 5. Conclusions

On the basis of the results of this study, premedication with steroids can reduce the recurrence of moderate and severe HSRs in patients who have a history of such reactions. However, the limitations in the current literature do not permit us to make a solid conclusion regarding the efficacy of such premedication, and future work is needed to garner more definitive evidence. However, based on the results, we suggest that premedication with steroids be administered to patients who have had previous moderate and severe HSRs. 

## Figures and Tables

**Figure 1 diagnostics-12-01673-f001:**
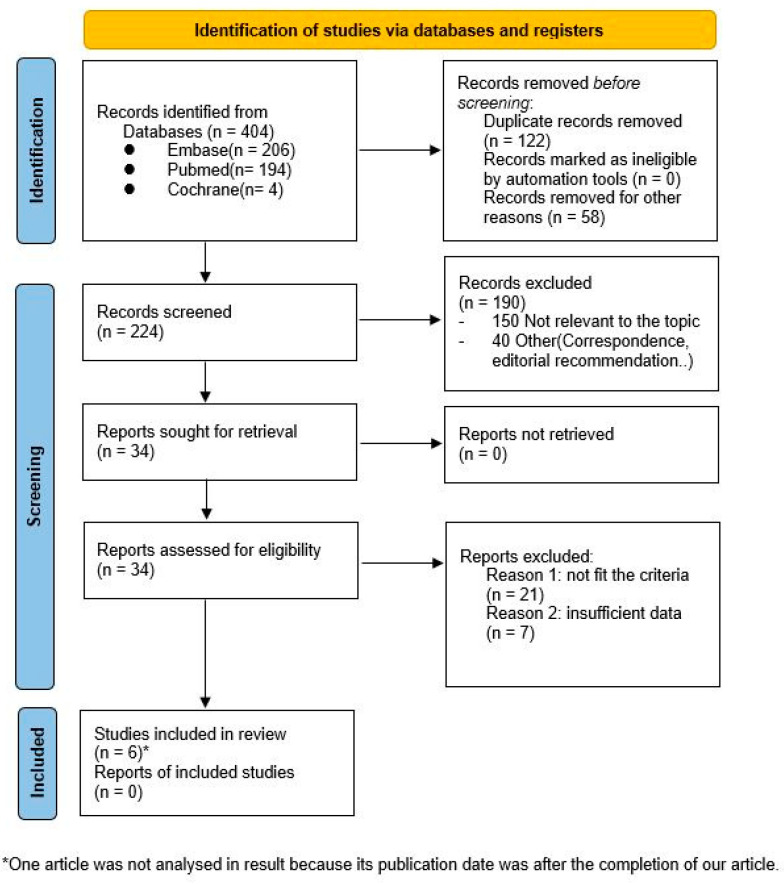
PRISMA 2020 flow diagram for new systematic reviews.

**Figure 2 diagnostics-12-01673-f002:**
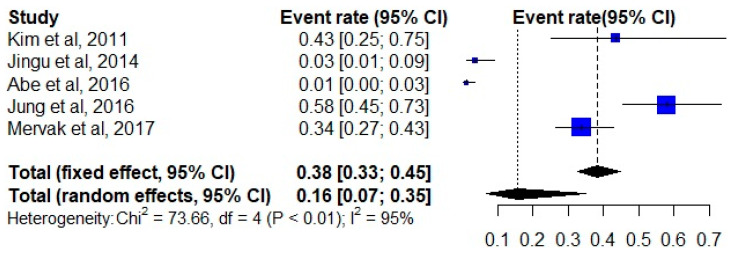
Forest plot of the initial event rate of hypersensitivity reactions in all included studies [26,27,28,29,30].

**Figure 3 diagnostics-12-01673-f003:**
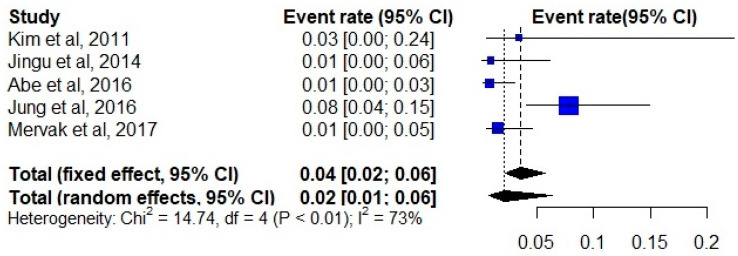
Forest plot of the event rate of hypersensitivity reactions in premedicated patients [26,27,28,29,30].

**Figure 4 diagnostics-12-01673-f004:**
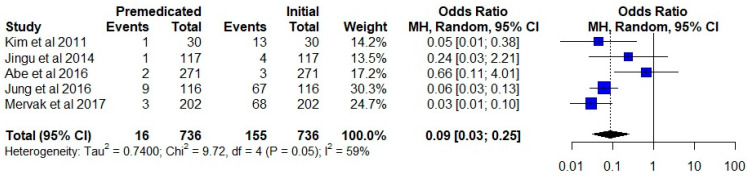
Comparison of the initial event rate versus the premedicated event rate in the same group of patients. The studies are shown with point estimates of the odds ratios and 95% confidence intervals [26,27,28,29,30].

**Table 1 diagnostics-12-01673-t001:** Characteristics of the five included studies.

Study	Country	Periods	Initial HSRs	Class of Contrast	Procedure	Number of Examinations	Age Mean(Range)	Sex Ratio (M/F)	Regimen of Steroid	Total Dose of Steroids (Prednisolone Equivalent)	Outcomes of Premedication
Kim et al., 2011 [29]	Korea	3 years	Moderate and Severe: 13	LOCM and IOCM	CT, Intervention	30	50.9 (21–77)	0.36	50 mg prednisolone or 40–50 mg methylprednisolone administered one to three times (13, 7, and 1 h prior to the procedure)	30~187.5 mg	Moderate and Severe: 1
Jingu et al., 2014 [26]	Japan	1 year and 6 months	Moderate: 4 Severe: 0	LOCM	CT	117	(15–87) ^b^	0.70	32 mg Methylprednisolone administered two times (12 and 2 h prior to the procedure) +/− Diphenhydramine chlorate 50 mg administered 1 h prior to the procedure	80 mg	Moderate: 1 Severe: 0
Abe et al., 2016 [30]	Japan	8 years and 9 months	Moderate: 2 Severe: 1	LOCM ^a^	CT	271	65.3 (24–93)	0.45	(1) 100–500 mg hydrocortisone + 10 mg chlorphenamine intravenously administered 0.5-1 h prior to the procedure (2) 30 mg prednisone orally administered the previous night and 3 h prior to the procedure + 30 mg fexofenadine administered 1 h prior the procedure ^d^	(1) 25–125 mg (2) 60 mg	Moderate: 1 Severe: 1
Jung et al., 2016 [28]	Korea	2 years	Moderate: 46 Severe: 21	LOCM	CT	116	— ^c^	— ^c^	40 mg methylprednisolone intravenously administered once or multiple times at least 1 h prior to the procedure + antihistamine (4 mg chlorpheniramine +/− 20 mg famotidine) intravenously administered 0.5–1 h prior to the procedure	59.61 mg ^e^	Moderate: 4 Severe: 5
Mervak et al., 2017 [27]	USA	9 years and 6 months	Moderate: 42 Severe: 26	LOCM and IOCM	CT	202	58 (11–86)	0.67	200 mg hydrocortisone and 50 mg diphenhydramine administered 9 and 1 h prior to the procedure	100 mg	Moderate: 1 Severe: 2

Abbreviations: HSRs: hypersensitivity reactions, LOCM: low-osmotic contrast media, IOCM: iso-osmotic contrast media. ^a^ The same contrast media were used; ^b^ Lack of mean age; ^c^ It was hard to extract the necessary information from the original article.; ^d^ An oral regimen was given to all patients who were able to take oral medication, and intravenous premedication was given to the patients who were unable to take oral medications and in urgent cases; ^e^ Mean systemic corticosteroids.

**Table 2 diagnostics-12-01673-t002:** Tool for evaluating the methodological quality of case reports and case series.

Domains	Leading Explanatory Questions
Selection	1. Does the patient(s) represent(s) the whole experience of the investigator (center) or is the selection method unclear to the extent that other patients with similar presentation may not have been reported?
Ascertainment	2. Was the exposure adequately ascertained? 3. Was the outcome adequately ascertained?
Causality	4. Were other alternative causes that may explain the observation ruled out? 5. Was there a challenge/rechallenge phenomenon? 6. Was there a dose–response effect? 7. Was follow-up long enough for outcomes to occur?
Reporting	8. Is the case(s) described with sufficient details to allow other investigators to replicate the research or to allow practitioners make inferences related to their own practice?

Questions 4, 5, and 6 are mostly relevant to cases of adverse drug events.

## Data Availability

The datasets generated or analyzed during the study are available from the corresponding author on reasonable request.

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
