# Peer review of "Pharmacological Prevention of Hypersensitivity Reactions Caused by Iodinated Contrast Media: A Systematic Review and Meta-Analysis"

_diagnostics, 2022, doi:10.3390/diagnostics12071673_

Round 1
Reviewer 1 Report
The manuscript entitled “Pharmacological prevention of hypersensitivity reactions caused by iodinated contrast media: A systematic review and 3 meta-analysis” by Chen Hsieh et al. presents a systematic review/meta-analysis of measures to prevent hypersensitivity reactions caused by iodinated contrast media.
The manuscript provides information on a subject for which a lot of literature has been published (387 listed in this manuscript). However, only five met the inclusion criteria set by the authors for this meta-analysis. This low rate of inclusion reflects the mediocre quality of studies describing hypersensitivity reactions caused by iodinated contrast media. The strength of evidence in the articles included in this analysis was relatively low, as all studies lacked control groups.
This lack of quality in available data is damaging to the present report as only partial conclusions can be drawn from the meta-analysis. Nevertheless, a conclusion emerges in the fact that premedication with steroids can reduce the recurrence of moderate and severe hypersensitivity reactions in patients who have a history of such reactions. Although this conclusion is limited and somehow intuitive, the present report has the merit to highlight the requirement for more thorough studies to assess the efficacy or premedication. It should be noted that these limitations are acknowledged by the authors in their conclusions.
The manuscript is well-written, easily understandable and supports the conclusions of the authors. Although this report is a meta-analysis, very few clues have been brought to the attention of the reader in term of mechanism leading to hypersensitivity. Side-effects of iodinate contrast agents were thought to be due to the excess iodine/iodide present in the formulation. However, recent studies have shown that some side-effects are in fact due to the direct action of the contrast agent to tissues. This has been demonstrated in the thyroid (J Nucl Med. 2018 Jan;59(1):121-126; J Clin Med. 2020 Jan 23;9(2):329.) and even though the relevance of the thyroid studies to hypersensitivity is far from demonstrated, this work should in my view be discussed in the present manuscript.
Author Response
Comment 1: Discuss the direct action of the iodinated contrast media(ICM) on tissues in the manuscript
RESP.: Thank you for this suggestion. It would have been interesting to explore this aspect. After reading the article discussing the effect of ICM on thyroid iodide content (J Nucl Med. 2018 Jan;59(1):121-126; J Clin Med. 2020 Jan 23;9(2):329.), it was interesting to know that the ICM induce a dramatic change in cellular proteins and lead to the modulation of 16 distinct pathways in thyroid tissue. The findings might be a new point of view for hypersensitivity in the use of ICM. However, this would not be possible in our study because we lacked detailed information to discuss the mechanism of hypersensitivity.

Reviewer 2 Report
This paper is good and well presented. The topic is of interest and the study limitations are acknowledged by the authors.
Author Response
Comment 1: This paper is good and well presented.
RESP.: Thank you for your words.

Reviewer 3 Report
REVIEW URL: https://susy.mdpi.com/user/review/review/27260955/Jf3tnNGj
THE ABBREVIATIONS USED IN MY REVIEW: F=figure (ex. F1 for “Figure 1”), io=instead of, L=line (followed by number), T (table, ex. T1 for “Table 1”)
MINOR CORRECTIONS AND SUGGESTIONS:
L37: “treatment with premedication” io “treatment of premedication”
L41: “is thus suggested”
L109: Name that “special tool”
TYPOS AND SUGGESTIONS:
L51:” ,” (the comma should have no blanc before)
L50 and the following: All references should be cited before the ending point (or comma) of the phrase such as “[2–4].” io “.[2–4]”; L57: “cells [8],” instead of “cells,[8]” etc.
L95 (and following): Use commas after each proposed number criteria (eq. “HSRs, ”)
L100: “were: ”
L124: “a significant heterogeneity” io “as significant heterogeneity”
L142: “males and females” io “male and female”
L150 (the 1st row of Table 1 [T1]): one may bold the text or shadow the 1st line of the table. The 7th column of T1 “number of patients”, the title of the 9th row (“Percentage of male patients”) and the parameter should be replaced by M:F sex ratio; the 10th column’s title “Regimen of steroids” and put a blanc before ”mg” in the cells of that column which lack it; title of the 11th column “Total dose of steroids”; in the whole 12th column put a blanc after “:” (the same in 4th column)
Use dots after the figure titles “Figure 2/3/4.” in their captions.
FINAL REMARK. In my opinion, any review process may also contain a dialog between the reviewer/editor and the authors (not only reviewer’s/editor’s monologue) and that is why all my suggested corrections and advices are debatable, so that the author(s) can send me a counter argument if he/she does not agree to any of my suggestions / critique and he/she wants to bring more clarifications to the issues invoked by me as a reviewer of this paper.
Author Response
Comment 1: Suggestions for grammar
RESP.: Thank you for pointing this out! All grammar errors have been revised accordingly.
Comment 2: The 7th column of T1 “number of patients”, the title of the 9th row (“Percentage of male patients”) and the parameter should be replaced by M:F sex ratio.
RESP.: Thank you for pointing this out! We have modified the percentage of male patients to the sex ratio for better readability. Besides, we have made a correction that the number of patients has been changed into the number of examinations. All included articles provided the number of patients receiving premedication before ICM exposure, but one article (Jingu et al, 2014) merely mentioned the number of examinations for analyzing the relationship between initial reaction and breakthrough reaction.

Reviewer 4 Report
I read with great interest the manuscript entitled “Pharmacological prevention of hypersensitivity reactions caused by iodinated contrast media: A systematic review and meta-analysis” submitted to Diagnostics MDPI.
The Authors review the efficacy of premedication in patients with previous moderate and severe adverse reactions to contrast media.
The paper is well written and structured.
I have some comments which may help improve the quality of the manuscript. These are the following:
Abstract:
Please modify the conclusions to make them more consistent with what emerged from the meta-analysis. The data are not robust enough to recommend premedication as reported in the “Conclusion” section of the manuscript.
Introduction:
Please add multidisciplinary strategies involving Allergists and Radiologists in the prevention of adverse reactions (see doi: 10.1016/j.jaip.2018.06.030).
Line 81: …ACR… please states the acronym.
Results:
Please explain if the article with doi: 10.1016/j.anai.2016.11.027 was excluded from the analysis. Otherwise, please include it.
Discussion:
Lines 229-230: …European Society of…. ACR Manual on… Please provide appropriate references.
Please discuss practical clinical implications of the results of meta-analysis.
Author Response
Response to Reviewer 4
- Comment 1: Modify the conclusions in the abstract to make them more consistent with what emerged from the meta-analysis
RESP.: As suggested by the reviewer, we have modified the conclusion in the abstract accordingly.
- Comment 2: Add multidisciplinary strategies involving Allergists and Radiologists in the prevention of adverse reactions (see DOI: 10.1016/j.jaip.2018.06.030).
.: Thank you! We have added the description in Lines 83-85. - Comment 3: Why the article with DOI: 10.1016/j.anai.2016.11.027 was excluded?
: The article (DOI: 10.1016/j.anai.2016.11.027) used severity-tailored prophylaxis in patients with a history of prior immediate hypersensitivity reaction . For the patient with mild reaction, only antihistamines were given according to the protocol. However, the outcome was demonstrated in a 100% stacked area chart it was hard to extract the actual number in patients who received premedication with antihistamines and steroids. Thus, the article was excluded. - Comment 4: Please provide appropriate references in Lines 229-230.
RESP..: We have added references accordingly.
- Comment 5: Discuss practical clinical implications of the results of a meta-analysis
: The benefits of premedication overweight the potential harms and costs (DOI: 10.1016/j.jaip.2018.06.030). We have modified the conclusion as “Premedication with corticosteroids prior to ICM administration is thus highly recommended in high-risk patients.”
In addition, all spelling and grammatical errors have been corrected accordingly.
